# Soft Computing Techniques for Appraisal of Potentially Toxic Elements from Jalandhar (Punjab), India

Vinod Kumar [1],*, Parveen Sihag [2], Ali Keshavarzi [3], Shevita Pandita [4] and Andrés Rodríguez-Seijo [5,6]

1 Department of Botany, Government Degree College, Ramban 182144, Jammu and Kashmir, India
2 Department of Civil Engineering, Shoolini University, Solan 173112, Himachal Pradesh, India; parveen12sihag@gmail.com
3 Laboratory of Remote Sensing and GIS, Department of Soil Science, University of Tehran, P.O. Box 4111, Karaj 31587-77871, Iran; alikeshavarzi@ut.ac.ir
4 Department of Botany, University of Jammu, Jammu 180006, Jammu and Kashmir, India; pandita.shweta@rediffmail.com
5 CIIMAR-UP, Terminal de Cruzeiros Do Porto de Leixões, Avenida General Norton de Matos, 4450-208 Matosinhos, Portugal; andresrodriguezseijo@hotmail.com
6 Biology Department, Faculty of Sciences, University of Porto, 4169-007 Porto, Portugal
* Correspondence: vinodverma507@gmail.com

**Abstract:** The contamination of potentially toxic elements (PTEs) in agricultural soils is a serious concern around the globe, and modelling approaches is imperative in order to determine the possible hazards linked with PTEs. These techniques accurately assess the PTEs in soil, which play a pivotal role in eliminating the weaknesses in determining PTEs in soils. This paper aims to predict the concentration of Cu, Co and Pb using neural networks (NNs) based on multilayer perceptron (MLP) and boosted regression trees (BT). Statistical performance estimation factors were rummage-sale to measure the performance of developed models. Comparison of the coefficient of correlation and root mean squared error suggest that MLP-established models perform better than BT-based models for predicting the concentration of Cu and Pb, whereas BT models perform better than MLP established models at predicting the concentration of Co.

**Keywords:** boosted regression trees (BT); ecological risk assessment; heavy metals; lead; multilayer perceptron (MLP); neural networks (NNs); soil carbon; phosphorus





## 1. Introduction

Contamination by potentially toxic elements (PTEs) is one of the key worldwide environmental concerns due to their implications on all kinds of environments, the food chain, soil organisms, and humans through direct or indirect exposure [1–4]. The increasing urbanization, historical and recent industrial/mining activities, military activities, and agricultural practices owing to the usage of organic or inorganic fertilizers and agrichemicals are some of the most important sources of soil contamination by PTEs (e.g., [4–10]). However, contamination by PTEs is not confined to the localized point where it occurs, since there is a diffuse and generally poorly studied contamination that affects all ecosystems, including groundwater systems. In this sense, information about the ability of PTEs to affect other nearby ecosystems, and thus to be more available to terrestrial organisms, is controlled by soil properties, such as pH, organic matter, exchange cations, Fe/Mn oxides, etc. [7–11]. Furthermore, to manage and regulate the metal contamination in soils is to be needed to assess the origin of contamination [12,13]. The diverse distribution of PTEs in soils, the widespread causes of contamination, and inappropriate monitoring knowledge are the key concerns for scientists in assessing the multi-source of PTEs in soils at a regional level; exploring suitable strategies to handle this problem is necessary imperative. Therefore, understanding all these aspects, modelling techniques are an imperative approach to assess the PTEs' origin and their interface with soil properties [14,15].

Different modelling techniques have emerged to help assess the origin of PTEs and their possible interaction with soil properties quickly and cost-effectively. Traditionally, geostatistical and mapping/GIS techniques have been used (e.g., [12,13]); however, linear regression techniques, comprising principal component analysis–multiple linear regression (PCA–MLR), and neural networks have been successfully applied in recent years for soil mapping and contamination prediction, since there are simple methods to source identification of soil contaminants that require relatively few samples and reduced workload [15–20]. Various researchers used these techniques to predict the concentration of PTEs; for example, Deng et al. [21] predicted the As, Pb, Cr, Cd and Hg content, with total Cd content and pH as covariates and $R^2$ varies from 0.109 to 0.456. Gholami et al. [22] also projected the content of Fe and Ni. Different researchers all over the globe have studied further, numerous explorations using machine learning techniques in diverse areas of environmental engineering [23–29]. Machine learning techniques such as neural networks (NNs), dependent on multilayer perceptron (MLP) and boosted regression trees (BT) and stepwise regression, can predict non-linear associations amid diverse parameters. These techniques give information about the significance of variables in the method that may assist in controlling the PTEs contamination in soil environs and diminishes the health perils of PTE revelation [30,31].

In India, extensive fertilizers, pesticides, and rapid growth of industrial and urban growth development have a great impact on soil contamination, but there is no suitable dataset regarding agricultural roadside soils and standard methodologies about modelling strategies [2,32,33]. To achieve this goal, we have applied different modeling approaches such as neural networks (NNs) based on multilayer perceptron (MLP) and boosted regression trees (BT) to predict the concentration of Pb, Co and Cu in roadside agricultural soils in Punjab (India). The outcomes of this study will help in controlling and regulating the pollution of PTEs in soil.

## 2. Materials and Methods

### 2.1. Study Area

The current area of assessment was District Jalandhar, Punjab. This District is located between two rivers; Beas and Sutlej. Loamy soil is mainly found in this area, which is due to the cool to warm climate based on sub-moist environments [34]. The geological substrate consists of alluvial deposits from the Quaternary age associated with 81 Indus allivial plains [35]. It makes up approximately 5.35% of the area of Punjab, and is one of the most highly populated areas of Punjab. The land consists of 90% agriculture, 7.4% non-agriculture and 2.1% forests. It has an extensive setup of roads and is a significant location for agriculture as well as textile and automobile spare part factories [32]. The climate of the study area is normally very hot during the summer season and very cold during the winter season, with rice and wheat as the main crops in the study area. The annual rainfall is about 600 mm year$^{-1}$. When samples were collected, the humidity was 77% and the temperature was 18 °C.

### 2.2. Soil Sampling and Analysis of Chemical Properties of Soil

Samples were collected at a depth of 0–15 cm from 70 locations in triplicates from Jalandhar (India) (Figure 1). Soil samples were air-dried and analyzed for various chemical parameters (pH, phosphorus (P), Ca, Mg, and organic carbon (C)) and PTEs (Co, Pb and Cu). Soil pH was measured by employing micro pH Analytical pH-meter in 1:2 soil/water extracts [36]. The Olsen method was applied to determine phosphorus [37], while calcium and magnesium were determined through EDTA titrations [38]. Walkley-Black wet oxidation method was used to determine C content [39]. The pseudototal Co, Cu and Pb contents were determined by acid digestion using aqua regia ($HNO_3$: HCl, 1:3 *v/v*). One gram of each oven-dried soil sample was digested with 12 mL of aqua regia and the solution was heated on the hot plate for 1–2 h. The digested samples were filtered and diluted with 50 mL of steam distillation water and used for analysis. Element analysis in

the extracts was resolute by atomic absorption spectrophotometer (AAS) (Model Agilent Technologies 200-Series AA). The limits of detection of the instrument are as follows: 5 µg L$^{-1}$ for Co, 1.2 µg L$^{-1}$ for Cu and 14 µg L$^{-1}$ for Pb. More details are given in Kumar et al. [33].

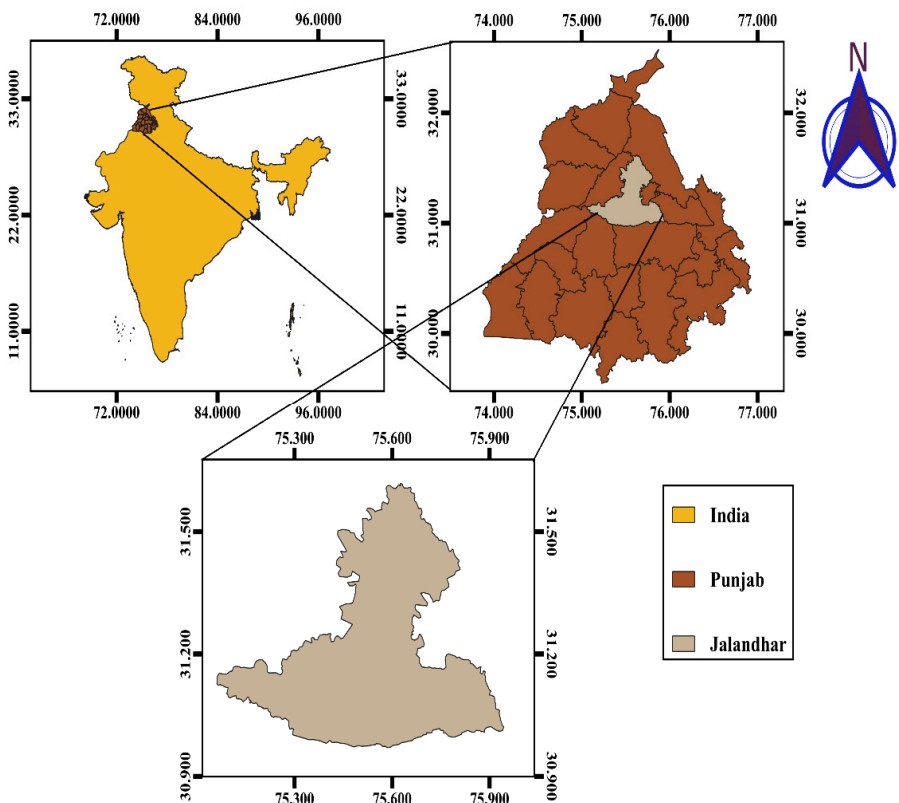

**Figure 1.** Study area showing different sites.

### 2.3. Stepwise Regression for Input Selection

The set of factors used for input vectors were pH, P, Ca, Mg and C. The corresponding outputs were Co, Cu and Pb. We accepted that the input vectors encompassed features that are expedient for influencing the output of PTEs. Afterwards, the variable selection method, i.e., the stepwise regression analysis, in which various combinations of input variables were tested together for input selection and pH and P were selected based on higher $R^2$ and lower residual mean square in Analysis of Variance (ANOVA) regression. Before analysis, data pre-processing was implemented using Sigma Plot (v. 12.0) and outliers were separated and then data dropped to 67 for further modeling processes. Subsequently, the data points were divided by the randomization technique. Data points were randomized and spited by Microsoft Excel software, and 70% of the data was selected to train the models and remaining 30% was used to test (15%) and validation (15%) of the developed models.

### 2.4. Modeling Techniques

Modeling approaches such as neural networks (NNs) based upon multilayer perceptron (MLP) and boosted regression trees (BT) are used in this paper for modeling of PTEs. The boosted regression trees (BT) are a group of two techniques; boosted and decision tree. Boosted was implemented with traditional techniques such as decision tree, M5P, support vector machine, etc. to improve performance. The basic principle of artificial neural network is human brain. The principally used design of NNs is serened of input, output and hidden layers known as MLP [38–41]. The details vis à vis modeling approaches were given in Shiri et al. [42,43] and Sihag et al. [15].

*2.5. Model Performance Assessing Parameters:*

For appraising the guessing aptitude of diverse approaches, the coefficient of correlation (CC) and root mean square error (RMSE) values were enumerated by using training and testing statistics. Elaborations are provided in earlier research by Sihag et al. [15].

$$\text{Coefficient of correlation} = \frac{n \sum E_{obs} E_{pred} - (\sum E_{obs})(\sum E_{pred})}{\sqrt{n(\sum E_{obs}{}^2) - (\sum E_{obs})^2} \sqrt{n(\sum E_{pred}{}^2) - (\sum E_{pred})^2}} \quad (1)$$

$$\text{Root mean square error} = \sqrt{\frac{1}{n}(\sum_{i=1}^{n}\left(E_{obs} - E_{pred}\right)^2} \quad (2)$$

where: $E_{obs}$ and $E_{pred}$ are experiential and prophesied values, and $n$ is number of observations. Figure 2 represents the overview of this paper.

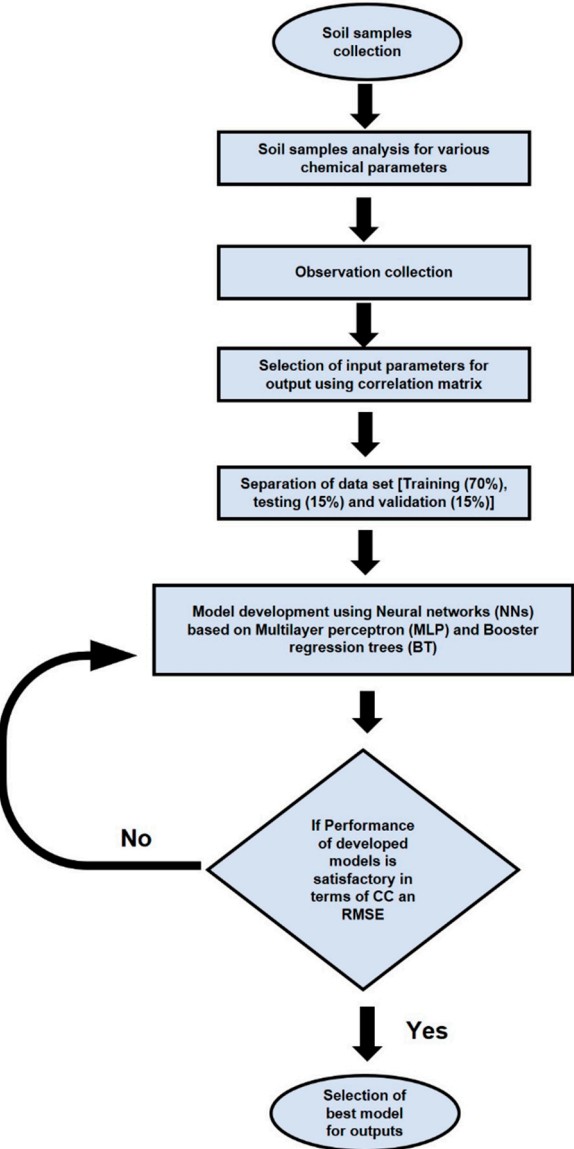

**Figure 2.** Flowchart of the work carried out in this paper.

## 3. Results and Discussion

The aim of the study was to evaluate the effectiveness of MLP- and BT-based models to predict the Co, Cu and Pb in the soil. Data used in this study were gathered from

field data. Figure 3 shows the correlation matrix of the dataset. Phosphorus and pH are positively correlated with Cu, Co and Pb, while Ca negatively correlates with these metals. Mg showed a moderate correlation with Cu, Co and Pb. Organic carbon also exhibits a moderate negative relationship with these elements. Figure 4 indicates the 3D plot of pH and P versus Co, Cu and Pb, respectively, dependent upon a set of three-dimensional points. These plots depict the relationship of phosphorus and pH with concentrations of Cu, Co and Pb. The effect of absolute variable can be examined using distinct plotting colours for the individual value of each variable. With the increasing concentration of phosphorus, the Co content increases, while the relationship of pH with Co shows that the concentration enhances with increasing values of pH, and maximum increase takes place at a pH value of 7.0. The maximum increase in Cu concentration occurs at phosphorus content 0.10 (mg kg$^{-1}$) and pH value of 7.3, while in Pb, maximum enhancement occurs at 0.20 (mg kg$^{-1}$) value of phosphorus and 7.0 value pH. Correlation coefficient and RMSE were used to assess the performance of developed models. The dataset consisted of 67 observations of studied variables where we can observe high levels of available P, and low levels of studied PTEs (Table 1).

**Table 1.** Descriptive statistics of studied variables (n = 67).

| Variables | Units | Mean | Minimum | Median | Maximum | SD | CV | Skewness |
|---|---|---|---|---|---|---|---|---|
| pH | | 7.76 | 6.70 | 7.88 | 8.80 | 0.41 | 0.05 | −0.21 |
| C | % | 3.66 | 1.78 | 3.51 | 6.70 | 0.99 | 0.27 | 0.80 |
| P | mg kg$^{-1}$ | 128.97 | 7.00 | 132.85 | 355.20 | 74.94 | 0.58 | 0.44 |
| Ca | meq 100g$^{-1}$ | 0.19 | 0.05 | 0.14 | 0.93 | 0.15 | 0.79 | 3.68 |
| Mg | meq 100g$^{-1}$ | 0.17 | 0.02 | 0.15 | 0.90 | 0.13 | 0.74 | 3.37 |
| Co | mg kg$^{-1}$ | 0.10 | 0.01 | 0.11 | 0.36 | 0.09 | 0.91 | 0.37 |
| Cu | mg kg$^{-1}$ | 0.51 | 0.01 | 0.19 | 12.79 | 1.99 | 3.89 | 5.77 |
| Pb | mg kg$^{-1}$ | 0.23 | 0.01 | 0.15 | 5.83 | 0.70 | 2.98 | 7.68 |

SD, Standard Deviation; C, Organic carbon; P, Available Phosphorus.

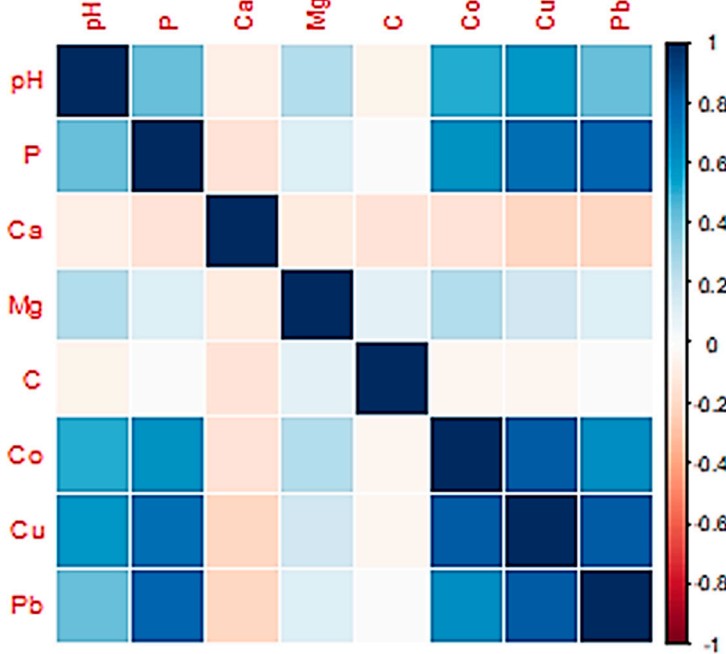

**Figure 3.** Correlation of input and target variables.

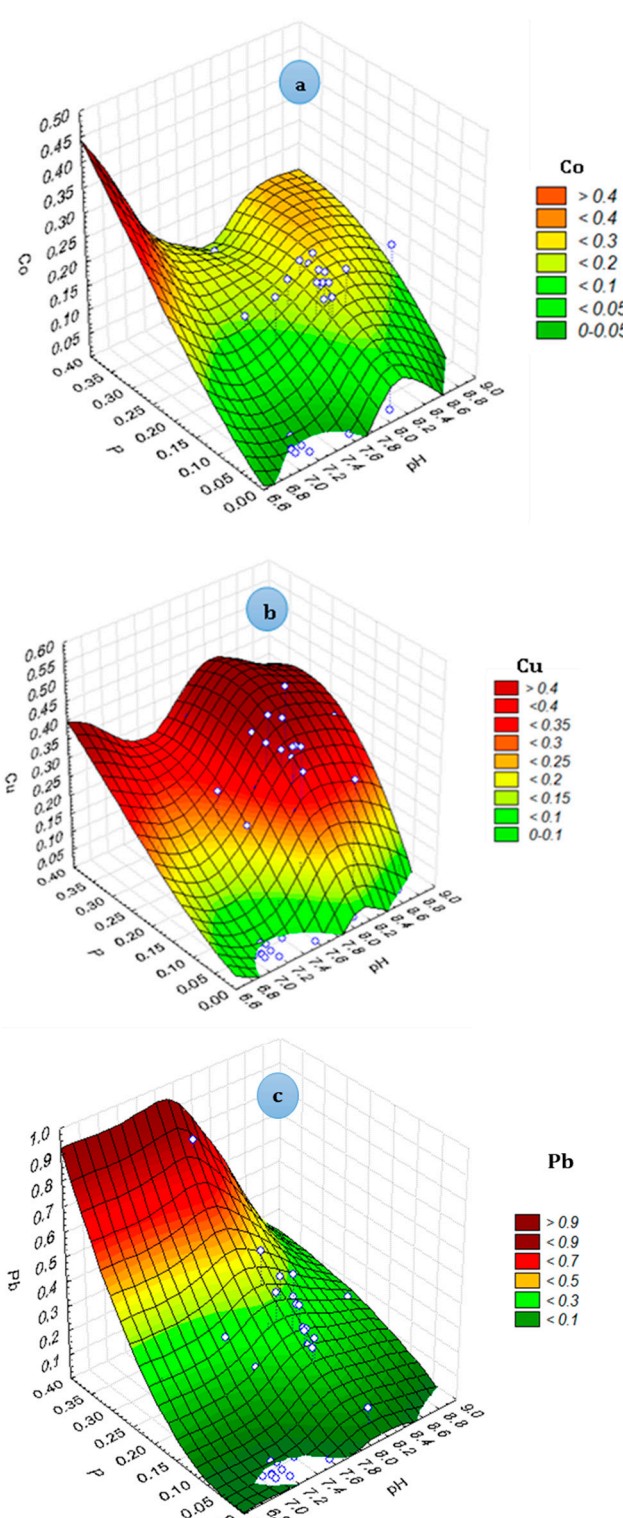

**Figure 4.** 3D surface plot of Co (**a**), Cu (**b**) and Pb (**c**) versus pH and P showing a three-dimensional functional relationship between designated dependent variable (Z = Co, Cu and Pb) and two independent variables (X = pH, Y = P).

### 3.1. Results of MLP-Based Models

The neural network models based upon MLP were executed by employing MATLAB software. The selection of input variables is the initial step in soft computing-based models' development. In this study, models were developed using pH, P, Ca, Mg and C variables.

Model development is a trial-and-error process. A larger dataset was used for model preparation, and other (smaller) datasets were used for model testing and validation. Different input combinations were used to be developed for predicting the Co, Cu and Pb. After analyzing Pearson's correlation matrix, five different models were developed. The different models are developed using a number of neurons in the hidden layer, and number of runs for output variables are Co, Cu and Pb. Figure 4 indicates the scatter plot of target and predicted values of Co using various MLP models. MLP-2-8-1 lies significantly closer to the line of perfect agreement (1:1) with lesser deviation. The model MLP-2-8-1 is best for predicting Co content, and the 2 signifies the number of input combinations (pH and phosphorus), while the 8 represents the number of neurons in a single hidden layer.

Table 2 indicates the values of coefficient of correlation and RMSE for all developed models for Co. Table 2 and Figure 5 suggest that that Model 4 that has the structure 2-8-1 is the best performing of all the developed models for all stages of model development with CC values in training (0.8547), testing (0.7186) and validation (0.5119), compared to the RMSE values obtained in training (0.0474), testing (0.0193) and validation (0.0060).

**Table 2.** Correlation coefficients and RMSE for different MLP topologies in all stages for Co.

| Models | | Training | | Testing | | Validation | |
|---|---|---|---|---|---|---|---|
| | Run | CC | RMSE | CC | RMSE | CC | RMSE |
| MLP 2-8-1 | 10 | 0.8676 | 0.0453 | 0.7226 | 0.0416 | 0.4026 | 0.0082 |
| MLP 2-5-1 | 10 | 0.8574 | 0.0469 | 0.7107 | 0.0162 | 0.4582 | 0.0070 |
| MLP 2-5-1 | 25 | 0.9075 | 0.0383 | 0.7645 | 0.0148 | 0.3138 | 0.0125 |
| MLP 2-8-1 | 25 | 0.8547 | 0.0474 | 0.7186 | 0.0193 | 0.5119 | 0.0060 |
| MLP 2-6-1 | 25 | 0.8511 | 0.0479 | 0.6662 | 0.0144 | 0.4769 | 0.0070 |

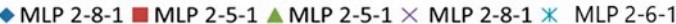

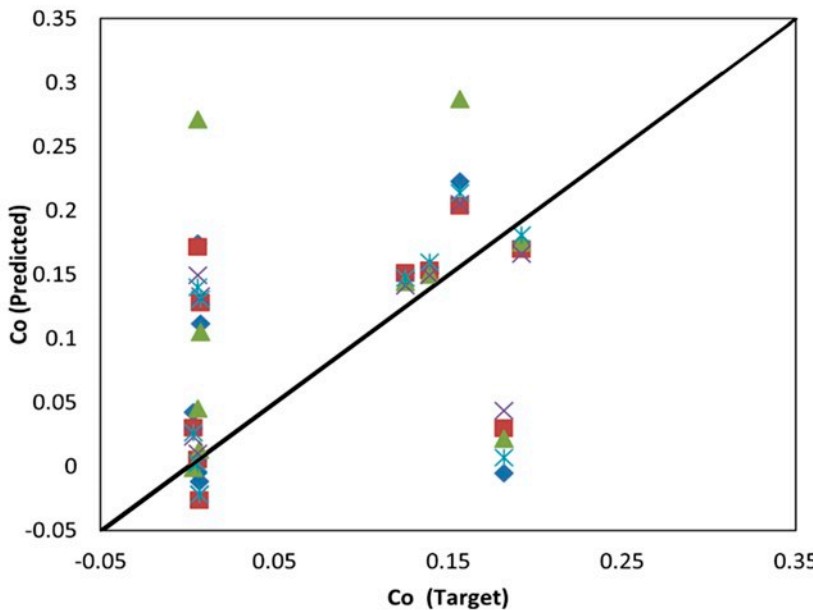

**Figure 5.** Scatter plot of target vs. output Co using different neural networks models dependent on MLP in validation stage.

Figure 6 shows the scatter plot of target vs. output Cu using various models for the validation stage. MLP-2-10-1 lies closer to the line of perfect agreement (1:1), while deviation is much less. Table 3 indicates the values of coefficient of correlation and RMSE for all developed models for Cu. This table suggests that Model 2, which has the structure 2-10-1

is the best performing of all the developed models for training and validation stages and shows comparable results in the testing stage with CC values in training (0.9488), testing (0.7366) and validation (0.8626), compared to the RMSE values obtained in training (0.0519), testing (0.0891) and validation (0.0943). The model MLP-2-10-1 is best for predicting the Cu concentration, and the 2 indicates the number of input variables used to build the model (pH and P) and the 10 indicated the number of neurons in a single hidden layer.

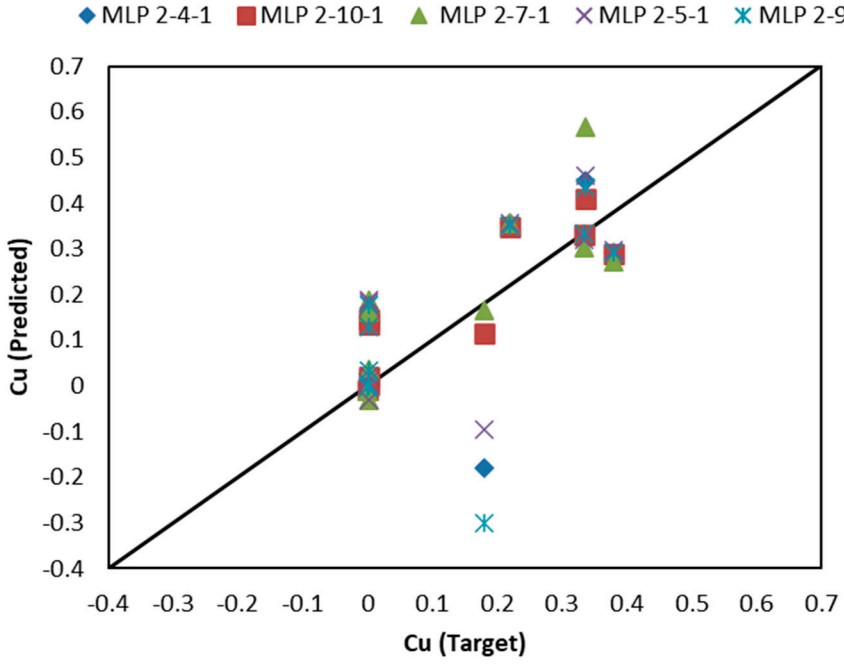

**Figure 6.** Scatter plot of target vs. output Cu employing numerous neural network models based on MLP in validation stage.

**Table 3.** Correlation coefficient and RMSE values for different MLP topologies in all stages for Cu.

| Models | Run | Training | | Testing | | Validation | |
|---|---|---|---|---|---|---|---|
| | | CC | RMSE | CC | RMSE | CC | RMSE |
| MLP 2-4-1 | 25 | 0.9407 | 0.0557 | 0.7420 | 0.0878 | 0.6635 | 0.1440 |
| MLP 2-10-1 | 25 | 0.9488 | 0.0519 | 0.7366 | 0.0891 | 0.8626 | 0.0943 |
| MLP 2-7-1 | 25 | 0.9331 | 0.0591 | 0.7554 | 0.0874 | 0.7883 | 0.0564 |
| MLP 2-5-1 | 25 | 0.9372 | 0.0573 | 0.7264 | 0.0901 | 0.6828 | 0.0903 |
| MLP 2-9-1 | 25 | 0.9463 | 0.0531 | 0.7393 | 0.0886 | 0.5610 | 0.0686 |

Figure 7 shows the scatter plot of target vs. output Pb using various models for the validation stage, and the MLP-2-10-1 model lies closer to the line of perfect agreement (1:1), while deviation is much less. Table 4 indicates the values of coefficient of correlation and RMSE for all the developed models for Pb, and this table suggests that that Model 4 with the structure 2-10-1 is the best performing of all the developed models for the training and validation stages, and shows comparable results in the testing stage with CC values in training (0.8562), testing (0.3706) and validation (0.7114), compared to RMSE values obtained in training (0.0231), testing (0.1071) and validation (0.0126). The model MLP-2-10-1 is the best for predicting Pb content, and the 2 signifies the number of input combinations (pH and phosphorus) and 10 is the number of neurons in a single hidden layer. From the inferences obtained using MLP models, we can say that metals concentration may nearly be appraised with these neural network models and it is comparatively easy to assess variables; it can be promising to identify metals that are detrimental to the feasibility of soils, both rapidly and economically [44]. Scatter plots revealed that inferences obtained through MLP-based

models of neural networks are acceptable for Cu, Co and Pb. Indeed, the determined models do not guarantee very high conformity amid assessments and amounts. However, this provides a useful method for assessing the eminence of soils [45]. El Badaoui [46], in their studies, applied a neural network approach based on MLP and multiple linear regression for predicting the concentration of Cu, Pb and Cr and inferred that NN models based on MLP are best predictors of the content of these metals with coefficients of determinations were 0.98 for Cu and 0.99 for Cr and Pb, respectively. Falamaki et al. [47] used various machine learning techniques in estimating the content of PTEs, for example, nickel, and concluded that our MLP-based NNs models better predict the content of PTEs in contrast with other models. Sihag et al. [15], while working on potentially toxic elements (Fe, Mn, Cu and Zn) in Neyshabur plain, Iran, applied different models such as NN-based MLP, M5 model tree (M5) and bagging approach (BM5P). They concluded that MLP models are the best predictors of Fe and Cu, while BM5P and M5P are appropriate models for predicting the Zn and Mn.

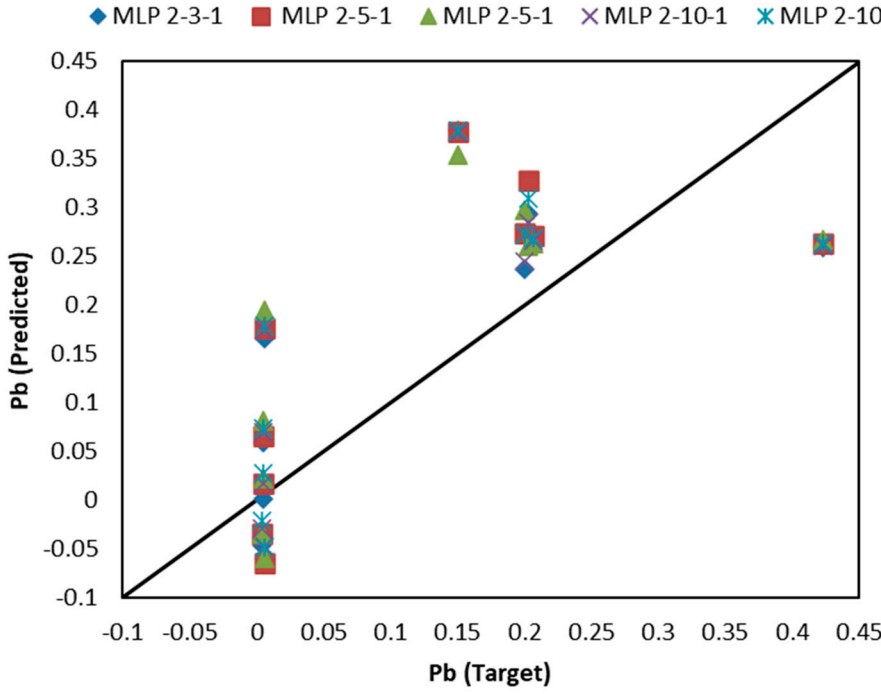

**Figure 7.** Scatter plot of target vs. predicted Pb using various neural networks models based upon MLP in validation stage.

**Table 4.** Correlation coefficients and RMSE values for different MLP topologies in all stages for Pb.

| Models | Run | Training | | Testing | | Validation | |
|---|---|---|---|---|---|---|---|
| | | CC | RMSE | CC | RMSE | CC | RMSE |
| MLP 2-3-1 | 25 | 0.8561 | 0.0938 | 0.3529 | 0.1078 | 0.7132 | 0.0128 |
| MLP 2-5-1 | 25 | 0.8465 | 0.0104 | 0.3490 | 0.1084 | 0.7113 | 0.0142 |
| MLP 2-5-1 | 10 | 0.8378 | 0.0243 | 0.3602 | 0.1052 | 0.7119 | 0.0129 |
| MLP 2-10-1 | 25 | 0.8562 | 0.0231 | 0.3706 | 0.1071 | 0.7114 | 0.0126 |
| MLP 2-10-1 | 10 | 0.8518 | 0.0066 | 0.3563 | 0.1079 | 0.7095 | 0.0137 |

### 3.2. Results of BT-Based Models

Excerpt of input variables is the initial step in developing BT-based models. In the present paper, the model was established using pH, P, Ca, Mg, and C. Model development is a similar process as that followed by the MLP-based model development. Figure 8 indicates the BT model-based tree graphs for Co, Cu and Pb, respectively. The BT regression trees

were obtained for Co, Cu and Pb using soil properties as forecasters. The root nodes of the regression tree in Co, spilt on phosphorus and pH, were also splitting variables into trees. It is assumed that lower phosphorus content is allied with greater Co retention, and pH is also an imperative variable in the retention of Co [48]. In the BT regression model of Cu and Pb, the root nodes of the regression tree also spilt on phosphorus and pH, and both these variables are imperative in maintaining Cu and Pb [49]. The agreement plots for target values and predicted values of Co, Cu and Pb are shown in Figure 9, respectively, using a validation dataset. CC and RMSE values of Co, Cu and Pb using BT-based models are listed in Table 5.

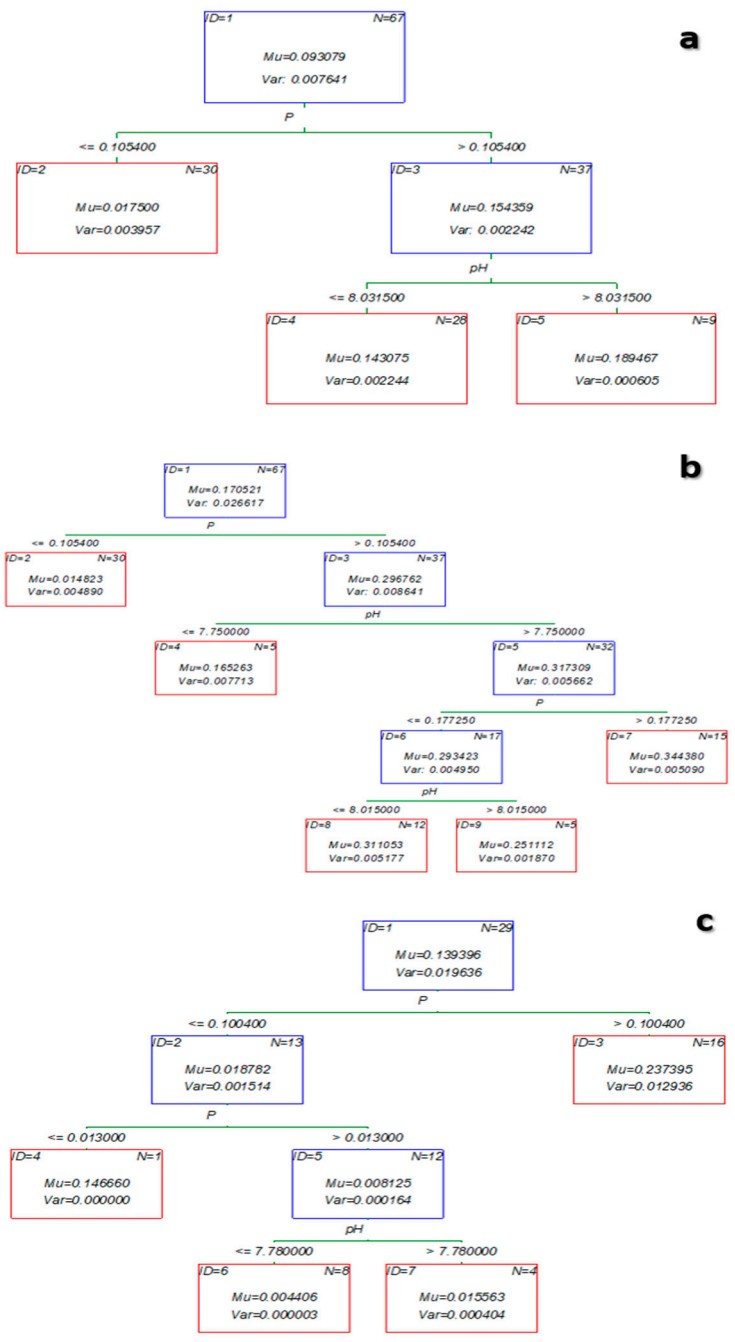

**Figure 8.** Boosted regression tree graph of Co (**a**), Cu (**b**) and Pb (**c**) (*Mu*, mean; *Var*, variance).

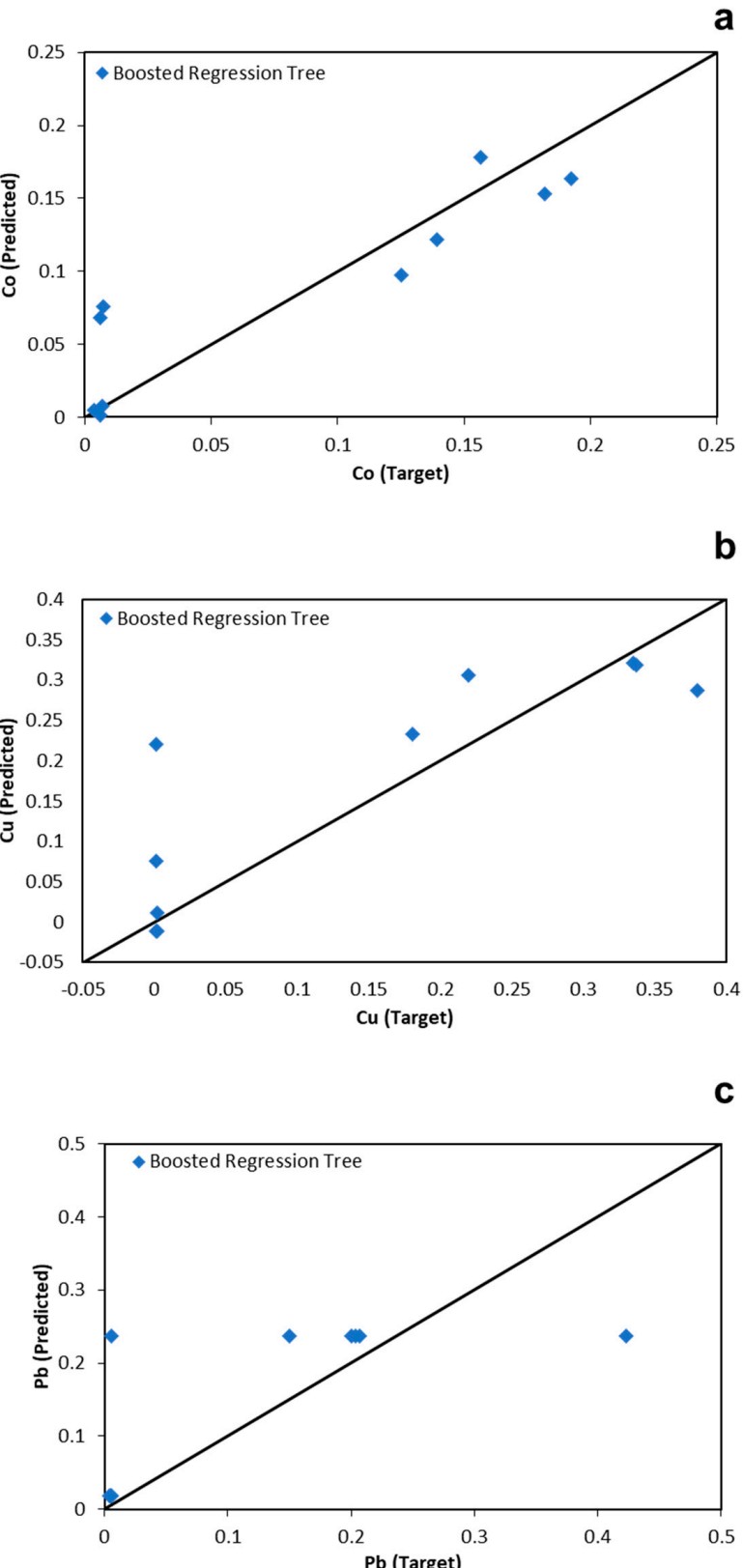

**Figure 9.** Scatter plot of target vs. predicted Co (**a**), Cu (**b**) and Pb (**c**) using the boosted regression tree method in validation stage.

**Table 5.** Correlation coefficients and RMSE values of Co, Cu and Pb using boosted regression tree method for all stages.

| Output | Training | | Testing | | Validation | |
|---|---|---|---|---|---|---|
| | CC | RMSE | CC | RMSE | CC | RMSE |
| Co | 0.9159 | 0.0376 | 0.7092 | 0.0455 | 0.9062 | 0.0343 |
| Cu | 0.9600 | 0.0478 | 0.8646 | 0.0755 | 0.8539 | 0.0854 |
| Pb | 0.8000 | 0.1175 | 0.6450 | 0.0791 | 0.7064 | 0.1000 |

Among the predicted level of metals, the highest CC values were found for Cu in training (0.960) and testing (0.8646), while in the validation of models, the highest CC values were found for Co (0.9062). The RMSE values were recorded as the highest for Pb in training (0.1175), testing (0.0791) and validation (0.1000). Wei et al. [50] used boosted regression, random forest and support vector machine techniques for predicting the PTE concentration, mainly arsenic, and found that among all applied models, boosted regression is the best model for predicting the arsenic concentration with RMSE (0.6007). Hu et al. [51], in their studies for predicting the PTEs (Zn, Cu, Cr, Ni, Hg, Cd, As, and Pb) concentrations, applied numerous modelling techniques such as boosted regression, random forest and generalized linear models. Among the applied models, random forest is the best, followed by boosted regression and generalized linear models for predicting the concentration of these PTEs with RMSE values Zn (0.067), Cu (0.059), Cr (0.033), Ni (0.044), Hg (0.021), Cd (0.229), As (0.103) and Pb (0.004).

### 3.3. Intercomparison of Applied Models

To better elucidate the model prophecy impact further, we applied the Taylor diagram [52]. The nearer the pentagram was to this line, the nearer the prophecy was to determine the Cu, Co and Pb concentration prediction [53]. The Taylor diagram is a polar graph in which the cosine of the angle amid the X-axis is the CC in Cu, Co and Pb of the model. The radial direction is the ratio of model to metals standard deviation. The grey arcs signify the RMSE normalized by the standard deviation for the apiece model [54]. Figure 10 shows the Taylor diagram for the comparison coefficient of correlation and RMSE in the validation stage for predicting Co, Cu and Pb using applied models. This suggests that MLP-established models perform better than BT-built models for predicting the Cu and Pb, whereas BT models perform better than MLP-based models in predicting the Co. Overall, the neural network models based on MLP are closer to the line and fit well, in contrast with other models for predicting the concentration of Cu, Co and Pb.

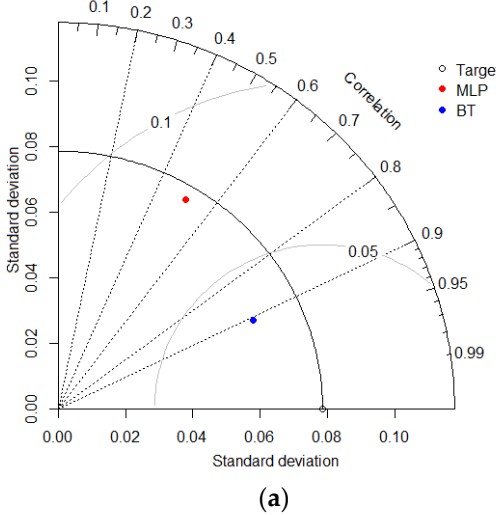

(**a**)

**Figure 10.** *Cont.*

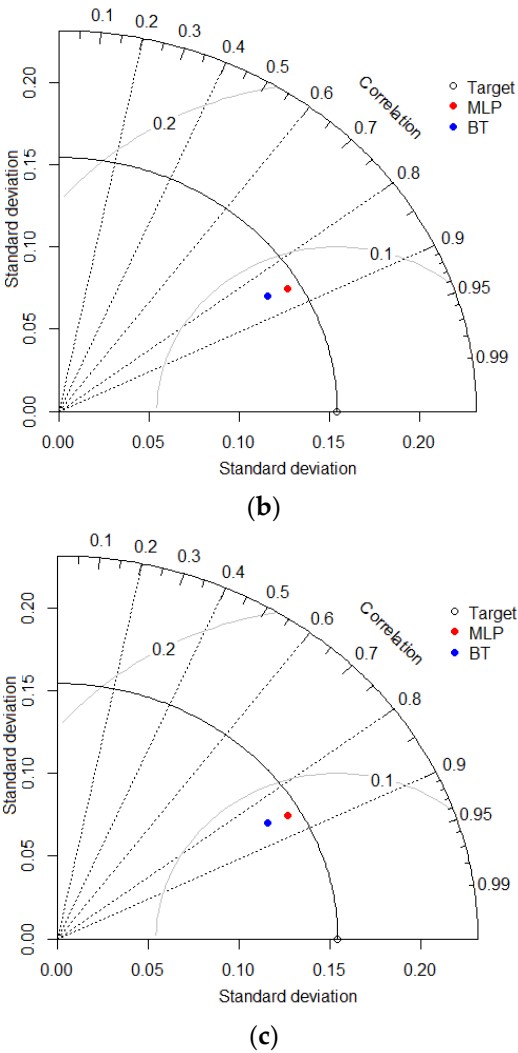

**Figure 10.** Taylor diagram for Co (**a**), Cu (**b**) and Pb (**c**) based models. The colored dots epitomize the models in the corresponding legend.

## 4. Conclusions

The contamination by PTEs is a severe concern for soils worldwide, and proper attention should be paid to overcoming this problem. Proper mitigation approaches are needed, and this study concludes that the concentration of Cu, Co and Pb in roadside soils was found less in contrast with Indian soil limits. We tried to predict the level of these elements using modelling techniques. We found that amid all the applied techniques, we can conclude that MLP-based models perform better than BT-based models for predicting the Cu level with RMSE (0.0519 to 0.0943) and CC (0.5619 to 0.9488) and Pb level with RMSE (0.0066 to 0.1084) and CC (0.3490 to 0.8562), while BT models perform better than MLP-established models in predicting the Co levels with RMSE (0.0343 to 0.0455) and CC (0.7092 to 0.9159), respectively. Further BT-based regression models indicate that pH and phosphorus are the imperative variables in the retention of Cu, Co and Pb in the soil. These findings were supported by Pearson's correlation analysis. Out of applied input soil variables in this study for model building, only phosphorus and pH exhibit a positive correlation with Cu, Co and Pb, and this may be why both these variables are imperative for the retention of Cu, Co and Co Pb in the soil of the studied region. The findings of the modelling techniques in the prophecy of Co, Cu and Pb helps ecological researchers to estimate the sites of effluence, causes and guidelines where the PTEs are disseminating. Planning appropriate environmental supremacy methods requires the mitigation of PTEs pollution in the environment. The present work provides significant

information about the predicting power of machine learning techniques for Co, Cu and Pb concentration prediction and the models in which all datasets are grouped into a single learning framework. With the greater recitals and valuable characteristics of the generalized models, the projected scheme was effectively established for a set of Cu, Co and Pb; it should be executed further to a bigger dataset to form a widespread model.

**Author Contributions:** Conceptualization, V.K., A.K.; methodology, A.K.; software, A.K., P.S.; validation, S.P., A.R.-S., V.K.; formal analysis, A.K., P.S.; investigation, V.K.; resources, A.K.; data curation, A.K., P.S.; writing—original draft preparation, V.K.; writing—review and editing, V.K., P.S., A.R.-S.; visualization, A.K., P.S.; supervision, V.K.; project administration and funding acquisition, V.K. All authors have read and agreed to the published version of the manuscript.

**Funding:** This research received no external funding.

**Institutional Review Board Statement:** Not applicable.

**Informed Consent Statement:** Not applicable.

**Data Availability Statement:** The data presented in this study are available on request from the corresponding author.

**Acknowledgments:** ARS would like to acknowledge the Foundation for Science and Technology (FCT) and CIIMAR (Grant Numbers UIDB/04423/2020, UIDP/04423/2020 and research contract CEECIND/03794/2017).

**Conflicts of Interest:** All authors declared that there is no conflict of interest.

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
