# Peer review of "Soft Computing Techniques for Appraisal of Potentially Toxic Elements from Jalandhar (Punjab), India"

_applsci, doi:10.3390/app11188362_

Round 1

Reviewer 1 Report

The paper is interesting as it deals with an important issue of agricultural soils, using machine learning techniques.

However, some methodological aspects and some results need some clarification. Specifically:

  1. There are some confusing changes in the names of the input variables:

In line 89 you write: “pHH2O, extractable phosphorus, Ca, Mg, and soil organic matter (SOM)”

In line 100: “pH, P, Ca, Mg and OM”

In table 1: “pH, OC, AP, Ca, Mg”

In fig.2: “pH, P, Ca, Mg, C”

Please choose one and keep it for all the paper.

  1. Paragraph 3.1

In line 104 you write that pH and P were selected as input variables. Indeed, results of fig.2 and fig.3 seems confirm that sentence. However, in 3.1 you also consider combinations of input variables that include organic carbon and Ca. Please explain this apparent contradiction.

How many times do you run each network? The results in tables 2, 3 and 4 are:

  • the unique results for a one-time run of each network?
  • The average results for n-time run of each network?
  • The best results for n-time run of each network?

What was the termination criteria?

The acronyms used for the different MLPs are definitely unclear. You can not have in table 2 and in figure 4 the same acronym for different neural networks.

Figure 5 and table 3 shows different acronyms for neural networks. Those used in table 3 are the same of table 2, while those used in fig.5 are different. One of these is wrong. Do you used combinations of input variables and network architectures that are the same of those used for Co or not?

The same as above for figure 6 and table 4.

  1. Paragraph 3.2

Again, it’s not clear what input variables were used and how they were selected.

Author Response

Reviewer #1

Question: The paper is interesting as it deals with an important issue of agricultural soils, using machine learning techniques.

However, some methodological aspects and some results need some clarification. Specifically:

There are some confusing changes in the names of the input variables:

In line 89 you write: “pHH2O, extractable phosphorus, Ca, Mg, and soil organic matter (SOM)”

In line 100: “pH, P, Ca, Mg and OM”

In table 1: “pH, OC, AP, Ca, Mg”

In fig.2: “pH, P, Ca, Mg, C”

Please choose one and keep it for all the paper.

Response: Dear reviewer #1, Thank you for your comments and suggestions to improve our paper. Following your suggestions, we use same symbols in throughout the manuscript.

Question: Paragraph 3.1

In line 104 you write that pH and P were selected as input variables. Indeed, results of fig.2 and fig.3 seems confirm that sentence. However, in 3.1 you also consider combinations of input variables that include organic carbon and Ca. Please explain this apparent contradiction.

Response: Thanks for your suggestion, as per your suggestion we checked all results again and now corrected it input variables are pH and P for the analysis.

Question: How many times do you run each network? The results in tables 2, 3 and 4 are:

the unique results for a one-time run of each network?

The average results for n-time run of each network?

The best results for n-time run of each network?

What was the termination criteria?

Response: Thanks for your suggestion, we used 10 and 25 runs and best results were achieved at 25 runs. Maximum value of CC is the termination criteria.

The acronyms used for the different MLPs are definitely unclear. You can not have in table 2 and in figure 4 the same acronym for different neural networks.

Figure 5 and table 3 shows different acronyms for neural networks. Those used in table 3 are the same of table 2, while those used in fig.5 are different. One of these is wrong. Do you used combinations of input variables and network architectures that are the same of those used for Co or not?

The same as above for figure 6 and table 4.

Response: Thanks for your suggestion, as your suggestion we corrected the acronyms for neural networks in all Tables according to Figures now it’s correct. We use pH and P as input parameters for the prediction of Cu, Co and Pb.

Paragraph 3.2

Again, it’s not clear what input variables were used and how they were selected.

Response: Thanks for your suggestion, as your suggestion we clearly mentioned now pH and P consider as the input variables for the analysis because pH and P has highest correlation with outputs.

Reviewer 2 Report

The manuscript is missing section 1 heading, I assume it to be Introduction. Further the introduction section is very poorly written. I recommend the authors to use more simple words and sentence structure and have more technical content in the manuscript. The introduction section consists of only two paragraphs of which first paragraph is dedicated to PTEs background and the second paragraphs has very few technical introductions about the research. The scientific research questions and research hypothesis is completely missing from the manuscript.

In section 2.1 first paragraph the study area is interpreted as current area, please try to be consistent with the terminologies. Further it is hard to follow the information about the study area from Figure 1. The fonts are too small and very inconsistent. Also, the figure captions are incomplete, Figure captions should provide all details present in the figure independent to the manuscript.

The authors have mentioned the prediction of PTEs. What are the spatiotemporal dimensions of the predictions and the collected sample?

Please provide more details of the models and the model parameters. It is hard to follow the methodology based on the details provided.  Please expand Materials and Methods section, the details provided are not sufficient.

There is no discussion of the observed results. Please provide the physical significance of the obtained results, its broader impact and intellectual merit.

No equation number is provided for the equations in section 2.5, and the fonts are very inconsistent.

Please make all the figures readable with consistency in the font sizes.

The conclusion section is not able to present the novelty of the research and its benefit. It seems to be a conclusion of case study with numbers in the conclusion.

Author Response

Reviewer #2

The manuscript is missing section 1 heading, I assume it to be Introduction. Further the introduction section is very poorly written. I recommend the authors to use more simple words and sentence structure and have more technical content in the manuscript. The introduction section consists of only two paragraphs of which first paragraph is dedicated to PTEs background and the second paragraphs has very few technical introductions about the research. The scientific research questions and research hypothesis is completely missing from the manuscript.

Response: Dear reviewer #2, thank you for your comments. The introduction section was reviewed and improved according to your comments and suggestions. References were updated.

In section 2.1 first paragraph the study area is interpreted as current area, please try to be consistent with the terminologies. Further it is hard to follow the information about the study area from Figure 1. The fonts are too small and very inconsistent. Also, the figure captions are incomplete, Figure captions should provide all details present in the figure independent to the manuscript.

Response: As per your suggestion we changed the study area map in revised manuscript.

The authors have mentioned the prediction of PTEs. What are the spatiotemporal dimensions of the predictions and the collected sample?

Please provide more details of the models and the model parameters. It is hard to follow the methodology based on the details provided.  Please expand Materials and Methods section, the details provided are not sufficient.

There is no discussion of the observed results. Please provide the physical significance of the obtained results, its broader impact and intellectual merit.

Response: Suggestions have been incorporated in the revised manuscript.

No equation number is provided for the equations in section 2.5, and the fonts are very inconsistent.

Response: As per your suggestion we included the equation numbers in section 2.5 and changed their font size now its consistent.

Please make all the figures readable with consistency in the font sizes.

Response: As per your suggestion we changed few figures now these figures are readable with consistency in the font sizes .

The conclusion section is not able to present the novelty of the research and its benefit. It seems to be a conclusion of case study with numbers in the conclusion.

Response: Conclusion has been improved in the revised version.

Reviewer 3 Report

Dear Authors,  

The results presented are important and should be published. However, my major concerns are listed below:

Lines 67-74:  The authors should explain the reason why Cu, Co ab Pb were determined and not some other potentially toxic elements.

Lines 77-78: The geographical coordinates of this area should be added.

Lines 95-97: The element analysis is very important for this study. Therefore, the method for Cu, Co and Pb determination should be described in more detail.

Line 100: The information about the normal distribution of selected variables should be added and the applied tests like ANOVA should be justified.

Line 117: The principles of the modeling techniques should be shortly described.

Overall, mine suggestion is that the manuscript would be acceptable with major revision. Substantial changes should be carried out before acceptance.

Good luck!

Author Response

Reviewer #3

Dear Authors, 

The results presented are important and should be published. However, my major concerns are listed below:

Lines 67-74:  The authors should explain the reason why Cu, Co ab Pb were determined and not some other potentially toxic elements.

Response: Dear reviewer #3, thank you so much for your comments and suggestions. Due to lack of facilities, only we can analyze these elements.

Lines 77-78: The geographical coordinates of this area should be added.

Response: As per your suggestion we included the geographical coordinates of the study area in Figure 1.

Lines 95-97: The element analysis is very important for this study. Therefore, the method for Cu, Co and Pb determination should be described in more detail.

Response: Suggested changes have incorporated.

Line 100: The information about the normal distribution of selected variables should be added and the applied tests like ANOVA should be justified.

Response: Dear Reviewer we have not applied ANOVA in this paper, and information about other applied statistics have been added.

Line 117: The principles of the modeling techniques should be shortly described.

Response: As per your suggestion we described the principles of the modeling techniques.

Overall, mine suggestion is that the manuscript would be acceptable with major revision. Substantial changes should be carried out before acceptance.

Good luck!

Response: Thank you so much for your suggestions to improve our paper.

Reviewer 4 Report

A kézirat áttekintése címe: Soft Computing Techniques for Appraisal of Potentially Toxic Elements

A mérgezÅ‘ elemek szennyezettsége miatt a szerzÅ‘k megfelelÅ‘ témát választottak. A károsodásokkal járó rehabilitációs feladatok kibocsátásai és lerakódási következményei egyre növekvÅ‘ világméretű problémát okoznak, különösen a posztindusztriális régiókban. A gépi tanulás segíthet a szennyezÅ‘dések elÅ‘rejelzésében.

A mérgezÅ‘ elemek felhalmozódásának, dúsításának és oldatszállításának vizsgálata nagyon fontos téma az egész ország számára. A nehézfémek mesterséges szennyezÅ‘dését emberi tevékenység okozza. A nehézfémek többnyire a termÅ‘talajban dúsulnak (halmozódnak fel), ahol a talaj kolloidjaihoz kötÅ‘dnek. A talaj pufferként működik egy bizonyos határig, a nehézfémek mozgósítása és megjelenése a talajoldatban a közeg pH -jától és víztartalmától függ.

The paper still has a few flaws, mainly structural and regarding its lengthy and wordy style it should be significantly changed, and it is specially true for the Materials and Methods and the Conclusion.

Although the paper is statistics oriented, these statistics have not contained units of measure. There are not adequate quantities without units. The title of the paper implies more Geography, spatial analysis and validation of spatial analysis maps. Therefore, I would expect a map for instance the positions of sample site at Republic of India and current state (Administrative division) where the sample sites had been.

The chapter Conclusion is insufficiently ‘conclusive’. Broader implementation of the results is really needed there.

General comment:

The title is too general. You should modify the title by adding geographic position e.g.: at District Jalandhar.

 You should modify the keywords with more specific words (e.g.): tillage, agricultural cultivation etc. There is not specific word as agriculture soil – what do the authors think about these terms?

Introduction
The introduction has sufficient size when it compared to the other chapters. However, the important citations of relevant articles from other countries are also required because based on this chapter it seems that the mentioned problem is a local one.

Material and method

The main problem with material methods is that the important information and simple insufficient construction of elementary description are missing. The meteorological, natural and agricultural descriptions are missing. What are the endemic species and planted species at this region?

What is the original soil in this region?

By following the authors description, the experiment of sample taking and sample preparing methods cannot be replied. How many was the quantity of samples? What is the typical soil at the sample sites? What was the classification of the soil type at sample site? (WRB and USDA soil classification). What is the description of soil profiles? What was the soil textural type? What was the particle size of distribution? How many was the clay % content? It is one of the most important parts when you write about soil nutrient content article. Where had been plots, what was physical Geography positions? Need one of table about physical soil properties as particle size distribution, texture, CaCO3% content and pH. The process of deposition, the toxic elements moving and the differences between O and A horizon properties. What were the background values of Toxic elements and nutrient? This part of manuscript determinate to understanding. These are a bit confusing for the readers.

Fig 1 It is simply unreadable, the foreign person does not know the locations. It needs to be significantly modified.

Result and discussion

The current value of the natural background information won't be evident for the researchers and unfortunately, this part of the chapters will not be understandable for everyone in this case. What was the starting values at the earlier times? Are there same both of places? Could you find a connection between landscape elements and contaminations? The big problem is that the result and discussion cannot discuss in perspective of soil science and spatial analysis. Therefore the model results are not understandable.

Map and figs and their captions do not contain units.
Most of Diagrams are not readable in this form, the font size is too small at special at Fig 7. and Fig 9. You have to remake these Diagrams.  The Tables have wrong positions and are randomly placed.

The references contain too little pieces of international examples.

Specific comments and questions:

Line (L.) 31-37: need more international references
L. 39 - 49: in this part the ecosystem services and aspect of toxic elements have to be mentioned.
L. 70 - 74: The validation of PTEs model is missing as aim in this part.
L 77 - 85. What is the climate in this region? How many is a yearly precipitation? How much is the altitude at sample area? What are the original and agricultural vegetations and the main species?
This part is deficient. There is no information about soils and there are no descriptions. What is the international classification of soils of sample sites? What is the soil physical properties? The author have to add table about base physical properties of sample area. How many is the background value of investigated toxic elements?
L. 86 -97: How much were the weights of soil sample? Were mixture of the samples came from the same sample site? Was it an average sample? This part is not clear.
L. 99 - 111 Have you done previously correlation test between the variables?
L. 102.-105 Were natural distribution of data sets?
L. 108 – 111. There are not enough points to validation, the best case is 0,15 x 70 = 10,5 pieces points. It is not clear what kind of validation method had been used to validation. E.g.:  k-fold cross-validation  or k-Fold Cross-Validation or Leave-one-out Cross-Validation or Leave-one-group-out Cross-Validation or Nested Cross-Validation or Time-series Cross-Validation or Wilcoxon signed-rank test or McNemar’s test or 5x2CV paired t-test or 5x2CV combined F test? This description is essential to repeating your work, It need to give exact specification.

  1. 117 One of workflow figure helps to better understanding the author’s work.
    L. 264. What would like to show at fig 7.? This figure form and quality is unreadable.
    L. 304 The font is too small at Fig 9., therefore this figure form and quality is unreadable.

    Conclusion
    Unlike the other chapters of the "paper" is way too short and does not directly flows from the results and goals. Secondly, how it could be implemented and compared for other land use areas? What sort of areas should be analyzed with that presented method? In order to use your results, list areas where your results may be potentially implemented or applied. The prime problems are the landscape  and soil physics pattern is not applied in the manuscript. When one of article writes about contamination, this contamination has got spatial distribution too. It is not enough to make statistical analysis without maps. The authors have to show the environmental background.

    My final opinion is that I am going to accept after the minor revision (corrections to minor methodological errors and text editing).

Author Response

Reviewer #4

The paper still has a few flaws, mainly structural and regarding its lengthy and wordy style it should be significantly changed, and it is specially true for the Materials and Methods and the Conclusion.

Although the paper is statistics oriented, these statistics have not contained units of measure. There are not adequate quantities without units. The title of the paper implies more Geography, spatial analysis and validation of spatial analysis maps. Therefore, I would expect a map for instance the positions of sample site at Republic of India and current state (Administrative division) where the sample sites had been.

The chapter Conclusion is insufficiently 'conclusive'. Broader implementation of the results is really needed there.

Response: All your suggested changes have been added in the manuscript.

General comment:

The title is too general. You should modify the title by adding geographic position e.g.: at District Jalandhar.

Response: As per your suggestion we included the District Jalandhar in title.

You should modify the keywords with more specific words (e.g.): tillage, agricultural cultivation etc. There is not specific word as agriculture soil – what do the authors think about these terms?

Response: As per your suggestion we included the potentially toxic elements in the keywords maximum soil samples were collected near road sides. So we removed agriculture soil etc.

Introduction
The introduction has sufficient size when it compared to the other chapters. However, the important citations of relevant articles from other countries are also required because based on this chapter it seems that the mentioned problem is a local one.

Response: Thank you for your comments. References were updated, and new references were added. However, the reviewer can see that the Introduction section has several papers from Europe or other countries different to India, including some review papers focused on agricultural areas, such as references 3, 4, 8, and 9.

Material and method

The main problem with material methods is that the important information and simple insufficient construction of elementary description are missing. The meteorological, natural and agricultural descriptions are missing. What are the endemic species and planted species at this region?

What is the original soil in this region?

Response: As per your suggestion we included the soil type in study area is a loamy soil.

By following the authors description, the experiment of sample taking and sample preparing methods cannot be replied. How many was the quantity of samples? What is the typical soil at the sample sites? What was the classification of the soil type at sample site? (WRB and USDA soil classification). What is the description of soil profiles? What was the soil textural type? What was the particle size of distribution? How many was the clay % content? It is one of the most important parts when you write about soil nutrient content article. Where had been plots, what was physical Geography positions? Need one of table about physical soil properties as particle size distribution, texture, CaCO3% content and pH. The process of deposition, the toxic elements moving and the differences between O and A horizon properties. What were the background values of Toxic elements and nutrient? This part of manuscript determinate to understanding. These are a bit confusing for the readers.

Response: Dear reviewer thanks for your constructive suggestions, but this experiment is already completed. In future we will take care of these.

Fig 1 It is simply unreadable, the foreign person does not know the locations. It needs to be significantly modified.

Response: As per your suggestion we changed the study area with more details now its readable and more informative.

Results and discussion

The current value of the natural background information won't be evident for the researchers and unfortunately, this part of the chapters will not be understandable for everyone in this case. What was the starting values at the earlier times? Are there same both of places? Could you find a connection between landscape elements and contaminations? The big problem is that the result and discussion cannot discuss in perspective of soil science and spatial analysis. Therefore the model results are not understandable.

Map and figs and their captions do not contain units.
Most of Diagrams are not readable in this form, the font size is too small at special at Fig 7. and Fig 9. You have to remake these Diagrams.  The Tables have wrong positions and are randomly placed.

Response: We rearranged the Tables and include new Taylor diagrams in the revised manuscript.

The references contain too little pieces of international examples.

Response: References were updated accordng your suggestions.

Specific comments and questions:

Line (L.) 31-37: need more international reference

39 - 49: in this part the ecosystem services and aspect of toxic elements have to be mentioned.

70 - 74: The validation of PTEs model is missing as aim in this part.

L 77 - 85. What is the climate in this region? How many is a yearly precipitation? How much is the altitude at sample area? What are the original and agricultural vegetations and the main species?

This part is deficient. There is no information about soils and there are no descriptions. What is the international classification of soils of sample sites? What is the soil physical properties? The author have to add table about base physical properties of sample area. How many is the background value of investigated toxic elements?

Response: Dear reviewer we have tried to add your suggestions but we have not analysed the physical properties and there is no background value of these metals in the área.

86 -97: How much were the weights of soil sample? Were mixture of the samples came from the same sample site? Was it an average sample? This part is not clear.

Response: 1 kg of soil composite samples in triplicates were collected from each site and used for analysis.

99 - 111 Have you done previously correlation test between the variables?

Response: Numerous times we have applied in various papers.

102.-105 Were natural distribution of data sets?

Response: Data is normally distributed.

108 – 111. There are not enough points to validation, the best case is 0,15 x 70 = 10,5 pieces points. It is not clear what kind of validation method had been used to validation. E.g.:  k-fold cross-validation  or k-Fold Cross-Validation or Leave-one-out Cross-Validation or Leave-one-group-out Cross-Validation or Nested Cross-Validation or Time-series Cross-Validation or Wilcoxon signed-rank test or McNemar's test or 5x2CV paired t-test or 5x2CV combined F test? This description is essential to repeating your work, It need to give exact specification.

Response: First of all we prepare all model using 10 cross fold validation and find optimum user define parameters and then appling these optimum user define parameters model was developed using training data set and testing and validation data set was used to check the developed model performance.

117 One of workflow figure helps to better understanding the author's work.

Response: we included the flowchart for the analysis in the revised versión.

264. What would like to show at fig 7.? This figure form and quality is unreadable.

L. 304 The font is too small at Fig 9., therefore this figure form and quality is unreadable.

Response: we included new Taylor Diagram Figures in revised manuscript.

Conclusion
Unlike the other chapters of the "paper" is way too short and does not directly flows from the results and goals. Secondly, how it could be implemented and compared for other land use areas? What sort of areas should be analyzed with that presented method? In order to use your results, list areas where your results may be potentially implemented or applied. The prime problems are the landscape  and soil physics pattern is not applied in the manuscript. When one of article writes about contamination, this contamination has got spatial distribution too. It is not enough to make statistical analysis without maps. The authors have to show the environmental background.

My final opinion is that I am going to accept after the minor revision (corrections to minor methodological errors and text editing).

Response: The conclusión section has been revised.

Round 2

Reviewer 2 Report

The authors have significantly improved the manuscript. It covers the important topic of PTEs appraisal. 

Author Response

Dear Reviewer #2

Thank you so much!

Reviewer 3 Report

Dear Authors,

In my opinion the manuscript could be now acceptable for publication.

Good luck!

Author Response

Dear reviewer #3

Thank you so much!

This manuscript is a resubmission of an earlier submission. The following is a list of the peer review reports and author responses from that submission.

Round 1

Reviewer 1 Report

c

The work is appropriate for the journal, the results are interesting, however there are some observations that must be addressed before the manuscript can be accepted.

ABSTRACT

The abstract should be concise and specific and consequently should be revised.

INTRODUCTION

It is necessary to make very clear the objective of the work to close the section of introductory concepts.

The introduction section seems a bit disconnected. You must add information to connect some paragraphs

MATERIAL AND METHODS

The methodological design is very clear and appropriate.

RESULTS

Standard deviations should be included. However, the resolution of the figures is not good.

DISCUSSION

In addition, the results should be more discussed respect other studies.

CONCLUSIONS

Conclusions should be reenforced considering the main findings of the study.

Author Response

Reviewer #1

Comment: The work is appropriate for the journal, the results are interesting, however there are some observations that must be addressed before the manuscript can be accepted.

ABSTRACT

The abstract should be concise and specific and consequently should be revised.

Action: Dear reviewer thanks for your constructive comments. The necessary changes have been made in the revised manuscript.

Comment: INTRODUCTION

It is necessary to make very clear the objective of the work to close the section of introductory concepts.

The introduction section seems a bit disconnected. You must add information to connect some paragraphs.

Action: Dear reviewer the suggested changes have been made in the introduction section.  Also, the objectives have been revised.

Comment: MATERIAL AND METHODS

The methodological design is very clear and appropriate.

Action: Thanks for your positive comment.

Comment: RESULTS

Standard deviations should be included. However, the resolution of the figures is not good

Action: The standard deviations are already provided in the Table 1. Resolution of figures have been improved in the revised manuscript.

Comment: DISCUSSION

In addition, the results should be more discussed respect other studies.

Action: Discussion has been elaborated by citing more references.

Comment: CONCLUSIONS

Conclusions should be reenforced considering the main findings of the study.

Action: Conclusion has been rewritten as per suggestions.

Reviewer 2 Report

It was not clear how the modeling results could be used for identifying source of PTEs in soils which I believe was mentioned as a goal of the research.

line 107, CC and RMSE, introduce them in full for the first time.

Figure 3, is Cu result missing? what dose the color mean? need to add more information in the caption.

Is neural networks (NNs) used in the study? it was mentioned in section 2.4 however.

Figure 4, the plot names were not explained, especially some names are exactly the same! The same applies to Figure 5-6.

More details should be included in the results and discussions. In fact, very little explanation was presented for 3.2  and 3.3. The author should not assume the reader understand the meaning of results presented. 

4. conclusion was not clear how the modeling results could be used. Couldn't understand line 186-189. 

Author Response

Reviewer #2

Comment: It was not clear how the modeling results could be used for identifying source of PTEs in soils which I believe was mentioned as a goal of the research. line 107, CC and RMSE, introduce them in full for the first time.

Action: Dear reviewer #2, thank you so much for your comments and suggestions. We updated our manuscript according to your comments. Overall, the manuscript has been improved and abbreviations have been defined in full form.

Comment: Figure 3, is Cu result missing? what dose the color mean? need to add more information in the caption.

Action: Figure 3 is provided and more information of plots have been added in the manuscript.

Comment: Is neural networks (NNs) used in the study? it was mentioned in section 2.4 however.

Action: NN based models on MLP are used in the study. More detail has been provided in detail in the revised manuscript.

Comment: Figure 4, the plot names were not explained, especially some names are exactly the same! The same applies to Figure 5-6

Action: Necessary changes have been improved in the figures.

Comment: More details should be included in the results and discussions. In fact, very little explanation was presented for 3.2  and 3.3. The author should not assume the reader understand the meaning of results presented.

Action: Detail has been added in the suggested sections.

Comment: conclusion was not clear how the modeling results could be used. Couldn't understand line 186-189.

Action: Conclusion section has also been improved and revised as per the suggestions.

Round 2

Reviewer 2 Report

Description of method choose to modeling the data need further clarification.  It seems MLP was part of the NN modeling in the results and discussion section (only seen from line 268) but they were paralleled in the introduction and method sections.

Figure 4 and Table 2-4, why some model names are the same? Why details of model was not presented? 

change all PH to pH.

How is the performance of the best model in this paper compared to that identified in other studies? The manuscript is lack of comparison with other studies in results and discussion section. otherwise, novelty should be pointed out in conclusion.

Author Response

Dear Reviewer,

Thank you so much for your comments and suggestions to help us to improve our paper. In addition to your comments, different typing error mistakes were reviewed. Please, see below a detailed list of the addressed questions.

Comment: Description of method choose to modeling the data need further clarification.  It seems MLP was part of the NN modeling in the results and discussion section (only seen from line 268) but they were paralleled in the introduction and method sections.

Action: Dear Reviewer thanks for your comment. Actually this was a typing error mistake. We applied NNs models based upon MLP in our study.

Comment: Figure 4 and Table 2-4, why some model names are the same? Why details of model was not presented? 

Action: Necessary changes have been added in the manuscript.

Comment: change all PH to pH.

Action: Changes have been made in the manuscript.

Comment: How is the performance of the best model in this paper compared to that identified in other studies? The manuscript is lack of comparison with other studies in results and discussion section. otherwise, novelty should be pointed out in conclusion

Action: Suggestions have been addressed in the revised manuscript. Comparison of present work with other related studies to modeling has been added.

Round 3

Reviewer 2 Report

I don't think the content in the Figures and Tables are well explained, for example, why some of the model name are the same.  Are them same models?

The information provided for the models are still limited. Basically, comparison were made between the modeling methods within this study. Comparison with other models for its potential implication in predicting the PTEs is still lacking.